# PARP1 recruits DNA translocases to restrain DNA replication and facilitate DNA repair

Yen-Chih Ho[1], Chen-Syun Ku[1], Siang-Sheng Tsai[1], Jia-Lin Shiu[1], Yi-Zhen Jiang[2], Hui Emmanuela Miriam[1], Han-Wen Zhang[1], Yen-Tzu Chen[3], Wen-Tai Chiu[4], Song-Bin Chang[1], Che-Hung Shen[5], Kyungjae Myung[6], Peter Chi[2,7], Hungjiun Liaw[1]*

1 Department of Life Sciences, National Cheng Kung University, Tainan City, Taiwan, 2 Institute of Biochemical Sciences, National Taiwan University, Taipei City, Taiwan, 3 Department of Public Health & Institute of Environmental and Occupational Health Sciences, College of Public Health, National Taiwan University, Taipei City, Taiwan, 4 Department of Biomedical Engineering, National Cheng Kung University, Tainan City, Taiwan, 5 National Institute of Cancer Research, National Health Research Institutes, Tainan City, Taiwan, 6 IBS Center for Genomic Integrity, UNIST-gil 50, Ulsan, Republic of Korea, 7 Institute of Biological Chemistry, Academia Sinica, Taipei City, Taiwan

* liawh@mail.ncku.edu.tw

**Data Availability Statement:** All relevant data are within the manuscript and its Supporting Information files.

## Abstract

Replication fork reversal which restrains DNA replication progression is an important protective mechanism in response to replication stress. PARP1 is recruited to stalled forks to restrain DNA replication. However, PARP1 has no helicase activity, and the mechanism through which PARP1 participates in DNA replication restraint remains unclear. Here, we found novel protein-protein interactions between PARP1 and DNA translocases, including HLTF, SHPRH, ZRANB3, and SMARCAL1, with HLTF showing the strongest interaction among these DNA translocases. Although HLTF and SHPRH share structural and functional similarity, it remains unclear whether SHPRH contains DNA translocase activity. We further identified the ability of SHPRH to restrain DNA replication upon replication stress, indicating that SHPRH itself could be a DNA translocase or a helper to facilitate DNA translocation. Although hydroxyurea (HU) and MMS induce different types of replication stress, they both induce common DNA replication restraint mechanisms independent of intra-S phase activation. Our results suggest that the PARP1 facilitates DNA translocase recruitment to damaged forks, preventing fork collapse and facilitating DNA repair.

## Author summary

Replication stress induces genomic instability and is associated with cancer development. PARP1 is not only involved in base-excision repair (BER), but also restrains replication fork progression upon replication stress. However, PARP1 has no helicase activities, and the mechanism through which PARP1 participates in the formation of reversed replication forks remains unclear. In the present study, we identified novel protein-protein interactions between PARP1 and DNA translocases, including HLTF, SHPRH, ZRANB3, and SMARCAL1, with HLTF showing the strongest interaction among these DNA translocases. This finding is particularly important because PARP1 inhibitors are effective against

**Funding:** This work was supported by the Ministry of Science and Technology, Taiwan [107-2311-B-006-003, 108-2321-B-006-018, 109-2326-B-006-003, 110-2326-B-006-004] to HJL. The funders had no role in study design, data collection and analysis, decision to publish, or preparation of the manuscript.

**Competing interests:** The authors have declared that no competing interests exist.

homologous recombination deficient cancers. Our study reveals an additional function of PARP1, which restrains replication progression through interaction with DNA translocases and provides an important protective mechanism in response to replication stress.

## Introduction

DNA lesions interfere with DNA replication, posing a threat to genome integrity [1,2]. The DNA alkylating agent, methyl methanesulfonate (MMS), generates N7-methylguanine, N3-methyladenine, and N3-methylguanine and is commonly used to study the mechanisms involved in DNA repair and post-replication repair [3–8]. MMS induced DNA methylation is primarily repaired by base excision repair (BER) mechanism [9]. N-methyl purine DNA glycosylase (MPG) recognizes and removes N-methylpurine [10], and the resulting apurinic site is cleaved by apurinic endonuclease 1 (APE1), leading to the formation of a single-strand break (SSB) [11]. SSBs are repaired by DNA polymerase β (Polβ) and DNA ligase III [12]. BER requires poly (ADP-ribose) polymerase I (PARP1), which catalyzes the formation of poly (ADP-ribose) (PAR) onto itself and target proteins [13,14]. PAR serves as a binding motif for recruiting BER-associated proteins. X-ray repair cross-complementing 1 (XRCC1) and Polβ are recruited by PARP1 through binding with the PAR moiety [15–19]. Mutations in PARP1 result in BER defects and the accumulation of SSBs.

MMS induces replication stress. X-shaped sister-chromatid junctions (SJCs) appear in yeast cells after MMS treatment, a process, known as the template switching subpathway of post-replication repair, which depends on the DNA translocase Rad5 and the DNA recombinase Rad51 [20]. Rad5 is a ubiquitin E3 ligase that catalyzes the polyubiquitin at K164 in proliferating cell nuclear antigen (PCNA) and possess helicase activity, promoting SCJs formation [8,21,22]. SJC is believed to be necessary to fill gaps caused by DNA lesions [23,24]. In higher eukaryotic cells, recent advances revealed that stalled replication forks are converted into reversed forks under replication stress conditions, which can be observed by electron microscopy [25]. Although SCJ structures have not yet been observed in higher eukaryotes, both SCJs and reversed forks feature annealing between sister strands. In mammalian cells, helicase-like transcription factor (HLTF) and SNF2 histone linker PHD RING helicase (SHPRH) are Rad5 orthologs and share both structural and functional similarities. Both HLTF and SHPRH are E3 ligases that promote the polyubiquitination of PCNA, similar to yeast Rad5 [26–35]. Currently, only HLTF and RAD5 have been identified to function as ATP-dependent DNA translocases and catalyze fork reversal upon replication stress [22,31]. The fork reversal activity has not yet been demonstrated for SHPRH. In addition to HLTF, other DNA translocases, including zinc finger RANBP2-type containing 3 (ZRANB3) and SWI/SNF-related, matrix-associated, actin-dependent regulator of chromatin, subfamily A-like 1 (SMARCAL1), can convert stalled forks into reversed forks [36–39]. ZRANB3 contains an NZF domain that interacts with K63-linked polyubiquitin chains [36,40]. SMARCAL1 interacts with single-strand DNA binding protein replication protein A (RPA) to localize to damaged forks [41–43]. The current model proposes that HLTF and SHPRH catalyze the polyubiquitination of PCNA, recruiting ZRANB3 through binding with the polyubiquitin moiety. HLTF, SHPRH, and ZRANB3 promote the annealing of sister strands, whereas SMARCAL1 promotes the annealing of parental strands. The coordination of these DNA translocases leads to the formation of reversed forks [44].

Recent studies reveal that PARP1 is not only involved in BER but is also involved in restraining replication fork progression and the formation of reversed forks [25,45–47]. However, PARP1 possesses no helicase, and the mechanism through which PARP1 participates in

DNA replication restraint remain unknown. In this study, we identified novel protein-protein interactions between PARP1 and DNA translocases, including HLTF, ZRANB3, SMARCAL1, and SHPRH. We further revealed that SHPRH restrains fork progression upon replication stress, indicating that SHPRH plays an important role in fork reversal. These recruitments not only restrain DNA replication but also facilitate DNA repair.

## Results

### PARP1 deletion reduces DNA translocase levels at damaged forks

Previous studies have shown that PARP1 restrains DNA replication upon replication stress. The depletion of PARP1 or treatment with the PARP1 inhibitor olaparib relieves replication restraint [25,45–47]. However, PARP1 has no helicase domains or DNA translocase activities, and the mechanism through which PARP1 restrains DNA replication is unknown. To address this question, we tested whether PARP1 is able to recruit DNA translocases, such as HLTF, SHPRH, ZRANB3, and SMARCAL1 to stalled forks. SHPRH is an HLTF homolog with similar structural and enzymatic activity [26–30]. However, to date, whether SHPRH catalyzes fork reversal similar to HLTF under replication stress conditions remains unclear. To further extend our understanding of the role played by SHPRH, we included SHPRH in this study. We generated a PARP1 knockout (PARP1-KO) T24 bladder cancer cells using the CRISPR-based gene-knockout strategy. PARP1-KO was confirmed by western blot (S1A Fig). The PARP1-KO cell line showed an elevated sister chromatid exchange (SCE) frequency, verifying the phenotype of PARP1-KO cells (S2A and S2B Fig). Interestingly, we also found that overexpression of DNA binding domain (DBD) of PARP1 resulted in a high SCE frequency (S2C–S2E Fig), suggesting that DBD has a dominant negative effect that competes with endogenous PARP1.

Previously, we have shown that PARP1 and HLTF does not reveal any foci formation similar to PCNA using confocal microscopy [48], it appears that the association of PARP1 or DNA translocases to replication forks cannot be accessed by confocal microscopy. In order to test whether these proteins associated with replication tracks, we performed a robust *in situ* analysis of protein interactions at DNA replication forks (SIRF), using proximity ligation assay (PLA) coupled with 5' ethylene-2'-deoxyuridine (EdU) click chemistry [49]. Replication tracks are labeled with EdU, which can be conjugated with biotin through the click reaction. Specific antibodies against biotin and the target protein can be used to detect associations between replication tracks and the target protein. The resulting PLA foci in the nucleus indicate the associations between target proteins and replication tracks. To validate the SIRF assay, we used a single anti-biotin antibody (biotin/-) or microtubule associated protein EB1 (biotin/EB1) to test the SIRF assay and both failed to show any PLA foci (S3A–S3C Fig). In contrast, two anti-biotin antibodies derived from mouse and rabbit (biotin (R)/biotin (M)), showed many PLA foci, ranging from 20 to more than 200 foci in the nuclei (S3A–S3C Fig). Approximately 70% of cells showed PLA foci, indicating that approximately 70% of cells are in S-phase. Using PCNA as a positive control revealed very similar results to biotin (R)/biotin (M) (S3A–S3C Fig).

We then tested whether DNA translocases were associated with replication tracks and whether MMS induced the recruitment of DNA translocases to damaged forks. The specificity of these antibodies used in the SIRF assay was tested in immunoblotting (S1A–S1E Fig). We treated wild-type and PARP1-KO T24 cells with 0.01% MMS for 1 hour to induce DNA damage. Approximately 40%-60% of mock-treated wild-type T24 cells contained between 1 and 10 HLTF, SHPRH, ZRANB3, and SMARCAL1 PLA foci (Figs 1A–1H and S4A–S4D), with more PLA foci observed for HLTF and SHPRH than for ZRANB3 and SMARCAL1, with

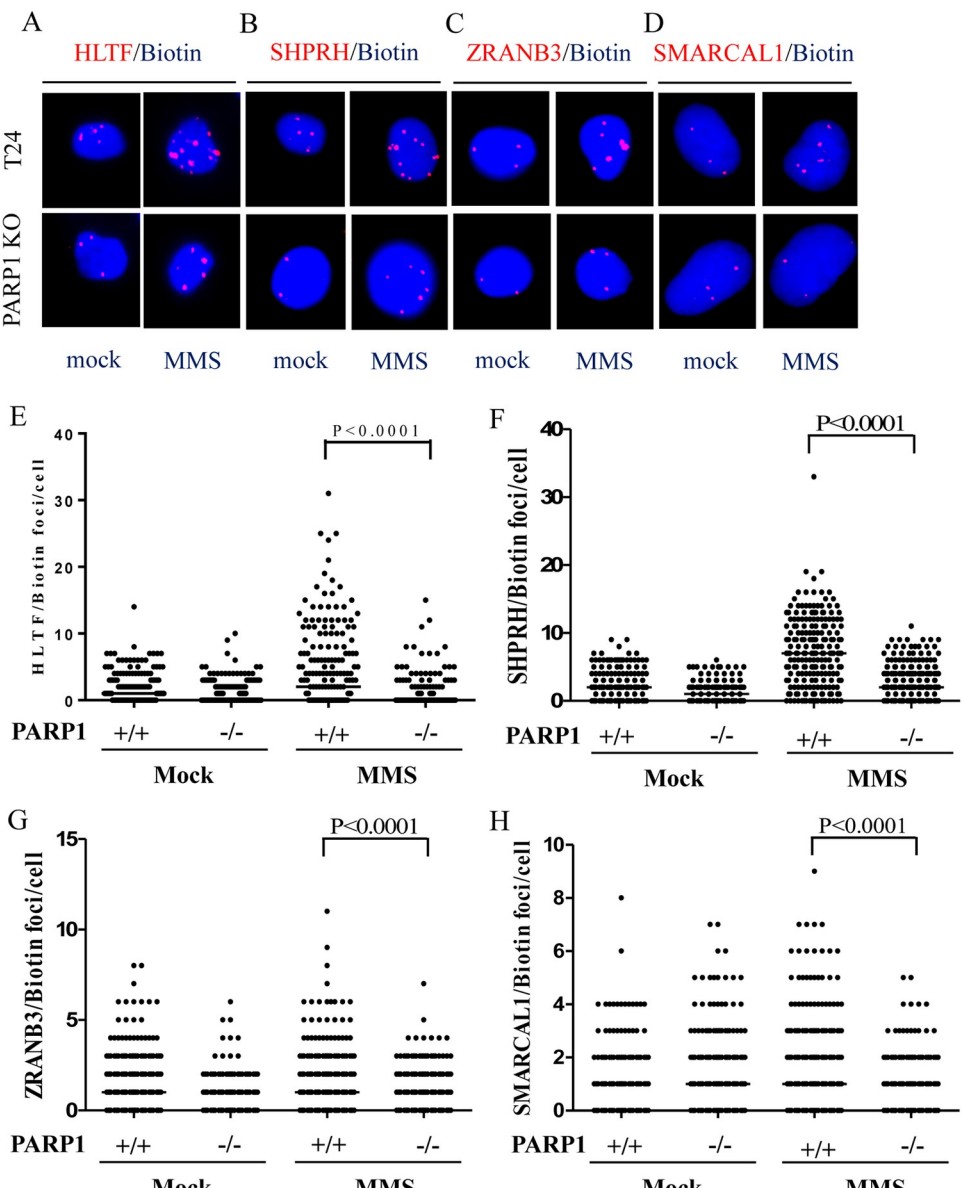

**Fig 1. The levels of DNA translocases are reduced at damaged replication forks in PARP1-KO cells.** (A)-(D) Representative images of HLTF, SHPRH, ZRANB3, and SMARCAL1 PLA foci, respectively, in wild-type and PARP1-KO T24 cells. Cells were treated with 0.01% MMS for 1 hour. The association of each protein with replication forks was determined by the SIRF assay. (E)-(H) Distributions of HLTF, SHPRH, ZRANB3, and SMARCAL1 PLA foci derived from a-d, respectively. At least 200 cells from each condition were measured. (Raw SIRF data in S1 Data).

SMARCAL1 displaying the fewest foci (Fig 1E–1H). Importantly, MMS treatment increased the number of PLA foci associated with all of these DNA translocases, with HLTF and SHPRH forming larger numbers of PLA foci than ZRANB3 and SMARCAL1 (Figs 1A–1H and S4A–S4D). UV treatment also induced an increase of PLA foci of all four DNA translocases (S5A–S5C Fig). By contrast, the numbers of PLA foci significantly decreased in PARP1-KO cells upon MMS treatment (Figs 1A–1H and S4A–S4D). We observed similar results using lentiviral packaged shRNA to deplete PARP1 expression (S6A–S6F and S7A–S7F Figs). The introduction of exogenous PARP1 into PARP1-KO cells restored the numbers of PLA foci

associated with all four DNA translocases (Figs 2A–2D and S8A–S8D and S9A–S9D). To test whether the PAR activity mediated by PARP1 facilitates DNA translocase recruitment, we introduced a PARP1-K893I mutant, which disrupts PAR activity [50,51]. We found that PARP1-K893I was not able to restore HLTF, SHPRH, SMARCAL1, or ZRANB3 PLA foci (Figs 2A–2D and S8A–S8D and S9A–S9D). Cells treated with PARP1 inhibitor olaparib also showed reduced HLTF and SHPRH PLA foci after MMS treatment (S10A–S10F Fig). Our results suggest that PARP1 facilitates the recruitment of DNA translocases to damaged forks in a PAR activity dependent manner. Both with and without MMS treatment, the numbers of PLA foci associated with these DNA translocases were fewer than those observed for PCNA (Figs 1A–1H and S3A–S3C) indicating that these DNA translocases are associated with some but not all active replication forks. These DNA translocase-associated replication forks may represent damaged forks arising from endogenous metabolites or exogenous MMS treatment.

## PARP1 associates with DNA translocases *in vivo*

To determine whether PARP1 associates with HLTF, SHPRH, ZRANB3, and SMARCAL1 DNA translocases, we performed coimmunoprecipitation assay with anti-PARP1 antibody. Proteins associated in the PARP1 complex can be determined by immunoblotting. Cells were either mock-treated or treated with 0.01% MMS for 1 hour. Since PARP1 and HLTF are able to bind DNA through their DBD domain and HIRAN domain, respectively [16,34,35], cell lysates were treated with DNase I or EtBr to rule out the possibility that DNA mediates these interactions. PARP1 was able to pull down all DNA translocases, HLTF, SHPRH, ZRANB3, and SMARCAL1 with HLTF showing the strongest interaction among these DNA translo-cases, and MMS treatment did not enhance these interactions. DNase I or EtBr treatment did not disrupt these interactions, suggesting these interactions are not mediated through DNA (Fig 3A).

Furthermore, we found that the interaction between PARP1 and SHPRH can be improved by overexpression of SHPRH. We transfected HEK293T cells with FLAG-tagged SHPRH and immunoprecipitated the FLAG-SHPRH complex using anti-FLAG sepharose beads. As shown in Fig 3B, FLAG-SHPRH was able to pull down PARP1. We further mapped the SHPRH domains that interact with PARP1. We generated the FLAG-tagged SHPRH (1–605) contain-ing the helicase ATP binding domain part 1 (HAB1) and H15 domain, SHPRH (606–1090) containing the PHD domain and helicase ATP binding domain second part (HAB2), and SHPRH (1090–1683) containing the RING domain and helicase C-terminal domain (HCT) using in-fusion cloning (S11 Fig). We found that SHPRH (1–605) and SHPRH (606–1090) were able to interact with PARP1 (Fig 3B). We have to note that SHPRH (606–1090) has lower expression levels compared to the other constructs. The C-terminal SHPRH (1090–1683) has marginal interaction with PARP1 (Fig 3B). Since the coimmunoprecipitation assay detects indirect and direct interactions between proteins, we cannot rule out the possibility that the interaction between PARP1 and SHPRH are mediated through other intermediates.

We further mapped the PARP1 domains that interact with each DNA translocase. We gen-erated GFP-tagged PARP1, PARP1 (1–374) containing the DNA binding domain, PARP1 (356–532) containing the BRCT domain, and PARP1 (529–1014) containing the catalytic domain (S12A Fig). We found that PARP1 (1–374) and PARP1 (356–532) interacts with HLTF, SHPRH, and SMARCAL1, but interacts with ZRANB3 weakly (S12B Fig). Instead, ZRANB3 showed stronger interaction with PARP1 (529–1014). Furthermore, PARP1 (529–1014) also showed weak and marginal interaction with SHPRH and SMARCAL1 (S12B Fig). We conclude that PARP1 interacts with these DNA translocases through multiple contacts.

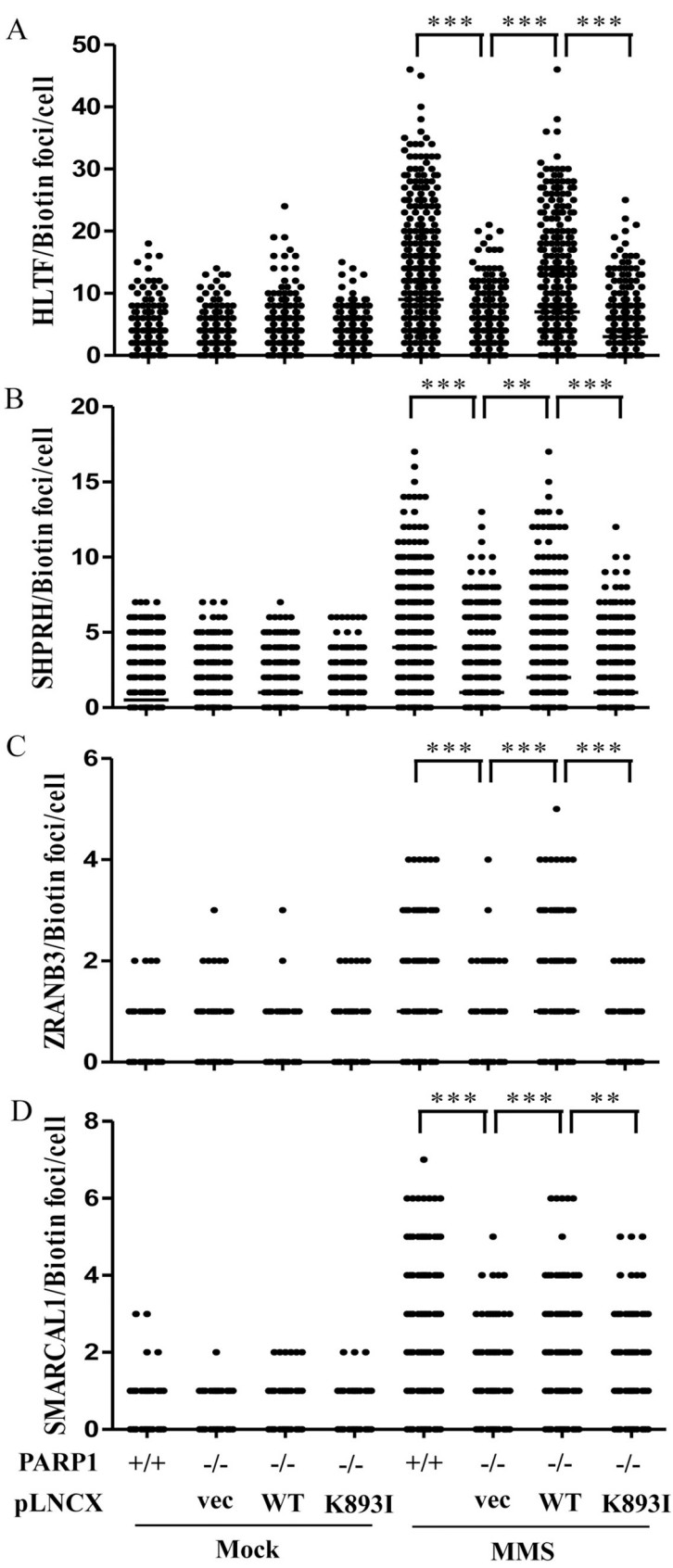

**Fig 2. DNA translocase levels at damaged replication forks are rescued by the introduction of PARP1 in PARP1-KO cells.** (A)-(D) Distribution of HLTF, SHPRH, ZRANB3, and SMARCAL1 PLA foci in PARP1-KO, PAPR1-rescue, and PARP1-K893I expressing T24 cells. The pLNCX vectors carrying wild-type PARP1 or PARP1-K893I mutant were packaged into retrovirus particles in GP2-293 cell line. PARP1-KO T24 cells were infected with these retroviruses to stably express wild-type PARP1 or PARP1-K893I mutant. Retrovirus carrying the empty vector (vec) was used as the control. Cells were treated with 0.01% MMS for 1 hour. The association of each protein with replication forks was determined by the SIRF assay. At least 200 cells from each condition were measured. (Raw SIRF data in S2 Data).

To test whether PARP1-DNA translocase interaction occurs *in vivo*, we examined protein-protein interactions using the PLA assay with specific antibodies against PARP1, HLTF, SHPRH, ZRANB3, and SMARCAL1. PARP1 interactions with DNA translocases appear as PLA foci. HLTF formed the most abundant PLA foci with PARP1, followed by SHPRH, SMARCAL1, and ZRANB3 (Figs 3C and 3D and S13). Interestingly, MMS treatment induced an increase in the numbers of HLTF/PARP1, SHPRH/PARP1, and ZRANB3/PARP1 PLA foci but only induced a marginal increase in SMARCAL1/PARP1 PLA foci (Figs 3C and 3D and S13). Similar to the result of MMS treatment, UV treatment also increased the numbers of HLTF/PARP1, SHPRH/PARP1, ZRANB3/PARP1, and SMARCAL1/PARP1 PLA foci (S14A–S14C Fig). Combined these results, our findings suggest that PARP1 interacts with DNA translocases and the interaction are not mediated through DNA.

## HLTF and SHPRH facilitate the loading of ZRANB3 to damaged replication forks

Previous studies have shown that HLTF and SHPRH are ubiquitin E3 ligases that catalyze the K63-linked polyubiquitination of PCNA [26,29,30]. ZRANB3 contains an NZF domain, which interacts with the K63-linked polyubiquitin chains on PCNA [36,40]. To test whether HLTF and SHPRH act synergistically to recruit ZRANB3 to damaged replication forks, we performed the SIRF assay. We found that MMS treatment induced an increase in ZRANB3 PLA foci in T24 cells (Fig 4A–4C), whereas the numbers of ZRANB3 PLA foci were reduced significantly in HLTF-KO/SHPRH-knockdown double-depleted cells (Fig 4A–4C). These results indicate that HLTF and SHPRH act synergistically to recruit ZRANB3 to damaged replication forks.

## DNA translocases restrain DNA replication upon HU-induced replication stress

Several lines of evidence have revealed that PARP1 and DNA translocases, including HLTF and ZRANB3, restrain DNA replication upon replication stress [25,35,39,46,47]. In the presence of low-dose hydroxyurea (HU) (50 μM) or camptothecin (50 nM), the depletion of PARP1, HLTF, or ZRANB3 results in longer replication tracks than in wild-type cells. We observed similar results in PARP1-KO, HLTF-KO, and ZRANB3-KO cells using a DNA fiber assay, with longer replication tracks upon 50 μM HU treatment compared to wild type cells (Fig 5A–5D). SHPRH depletion also exhibited longer replication tracks upon HU treatment compared to wild type cells (Fig 5E). To the best of our knowledge, this represents the first evidence that SHPRH restrains DNA replication under conditions of replication stress. We have to note that currently, there is no evidence presented that SHPRH contains DNA translocase activity, which contributes to fork restraining. It requires further biochemical study to verify its DNA translocase activity. Alternatively, SHPRH could serve as a helper or scaffold protein to facilitate fork reversal. It awaits further studies in the future.

Although HLTF and SHPRH share structural and functional similarities, the depletion of each gene alone was able to relieve replication restraint under conditions of replication stress.

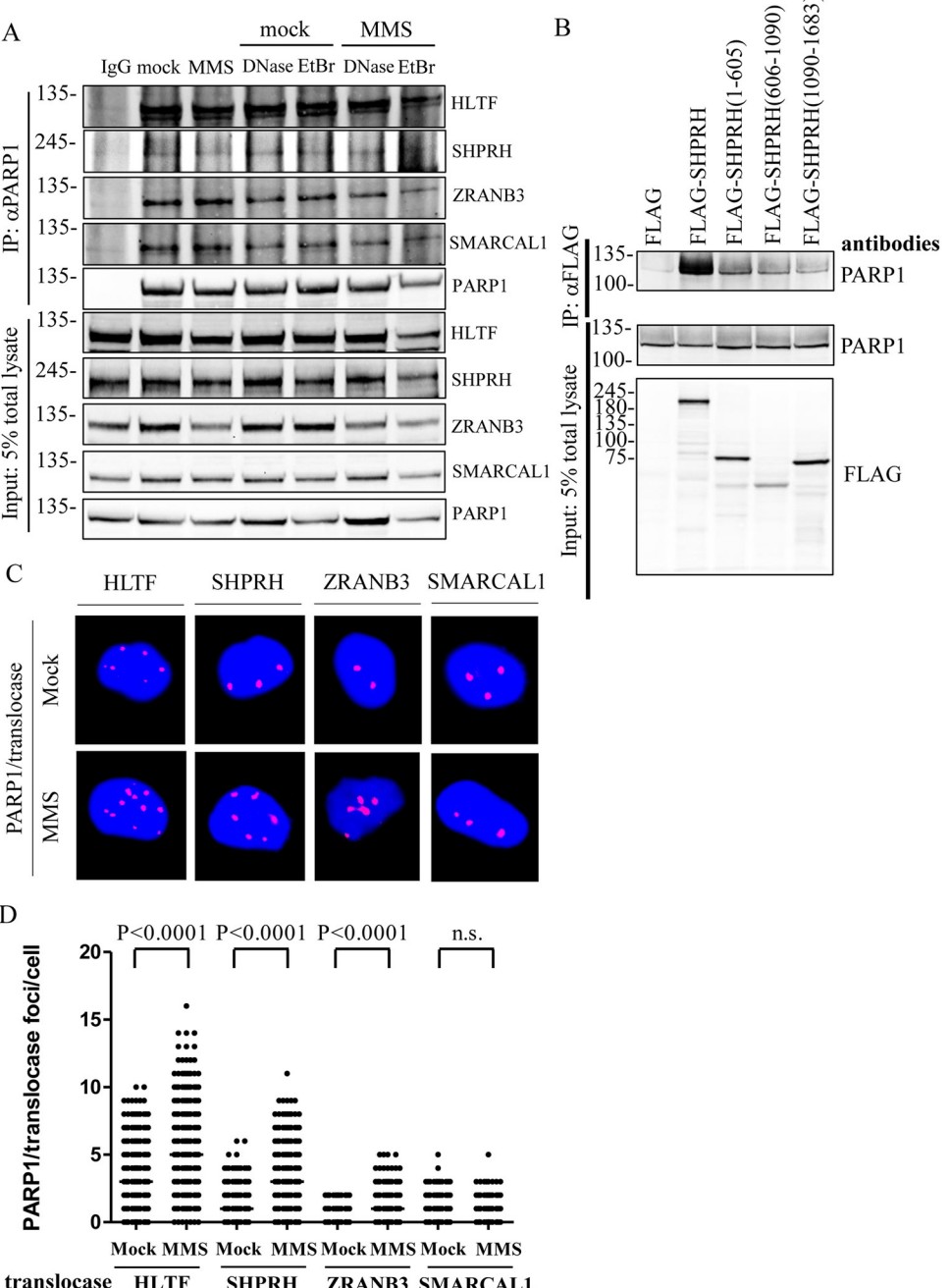

**Fig 3. PARP1 interacts with DNA translocases *in vivo*.** (A) The endogenous protein–protein interaction between PARP1 and translocases. Cells were treated with either mock or 0.01% MMS for 1 hour. Cell lysates were treated with either 100 μg/ml DNase I or 50 μg/ml EtBr as indicated. The endogenous PARP1 was immunoprecipitated with an anti-PARP1 antibody and the pulldown translocases were detected with specific antibodies as indicated. The non-specific mouse IgG was used as a negative control. Input represents 5% of total cell lysates. (B) The N-terminal domain of SHPRH interacts with PARP1. HEK293T cells were transfected with various FLAG–SHPRH constructs. The FLAG-SHPRH fusion proteins were immunoprecipitated with anti-FLAG sepharose beads, and the immunoprecipitates were then subjected to immunoblotting analysis. Input represents 5% of total cell lysates. (C) Representative images of HLTF, SHPRH, ZRANB3, and SMARCAL1/PARP1-PLA foci in T24 cells, respectively. Cells were treated with 0.01% MMS for 1 hour. The association of each translocase with PARP1 was determined by the PLA assay. (D) Distributions of HLTF, SHPRH, ZRANB3, and SMARCAL1/PARP1-PLA foci derived from C. At least 200 cells from each condition were measured. (Raw PLA data in S3 Data).

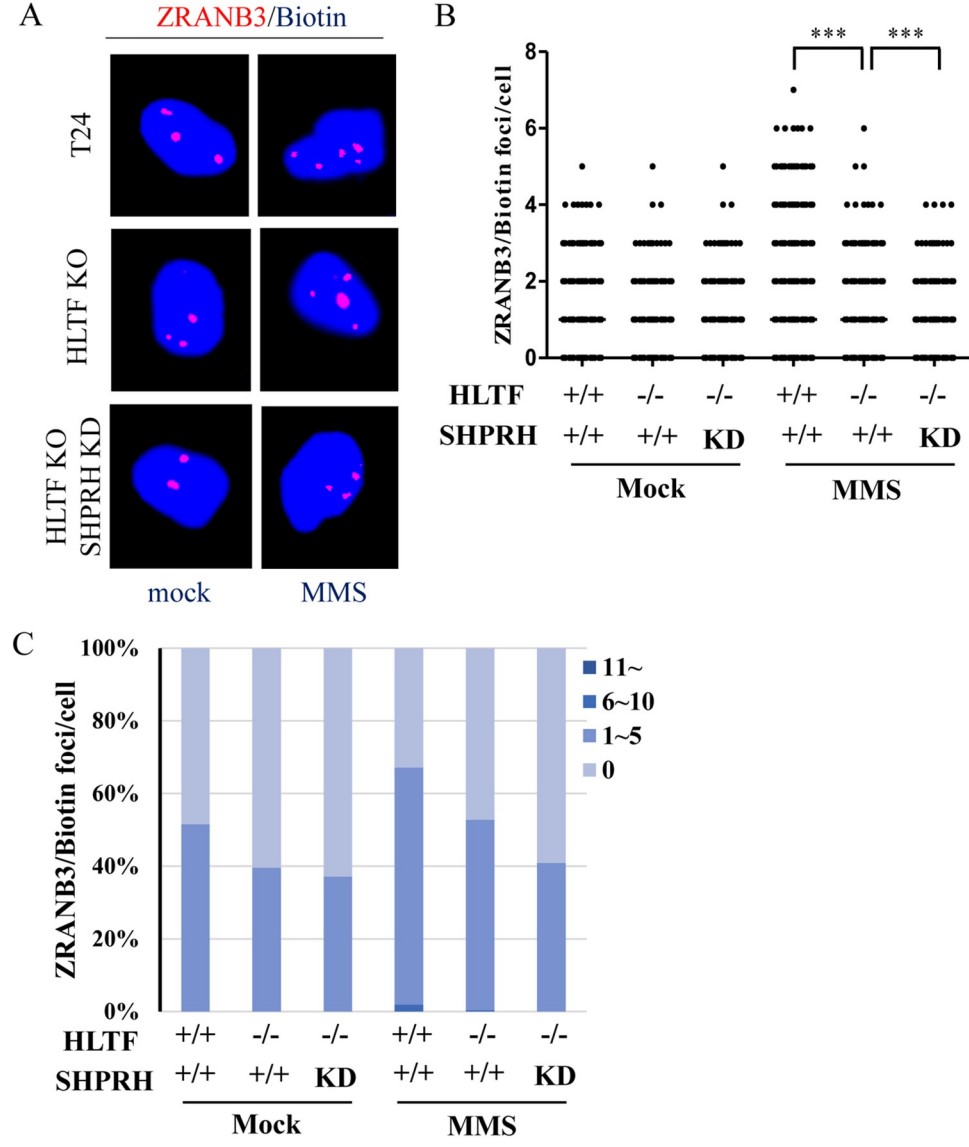

**Fig 4. ZRANB3 levels are reduced at damaged replication forks in HLTF/SHPRH double-depleted T24 cells.** (A) Representative images of ZRANB3 PLA foci in wild type, HLTF-KO, and HLTF/SHPRH double-depleted T24 cells. Cells were treated with 0.01% MMS for 1 hour. The association of ZRANB3 with replication forks was determined by the SIRF assay. (B) Distribution of ZRANB3 PLA foci derived from a. At least 200 cells from each condition were measured. (C) The number of PLA foci was classified into four groups: 0, 1–5, 6–10, and >11 foci, and the distributions of each group are shown in the plot. (Raw SIRF data in S4 Data).

To verify these results, we used an alternative approach in which active replication tracks were labeled with fluorescent Cy5, and the intensity of Cy5 correlates with the length of replication tracks. We labeled cells with EdU for 15 minutes, followed by treatment with 50 μM HU for 1 hour (S15A Fig). EdU incorporated into DNA was then conjugated with a fluorescent probe Cy5 using the click reaction, and fluorescence intensity was measured by fluorescence microscopy. Stronger Cy5 intensity indicates longer replication tracks. Using this alternative method, we are able to verify the results obtained in the DNA fiber assay. Wild type, PARP1-KO, HLTF-KO, ZRANB3-KO, and SHPRH-KD cells showed similar Cy5 intensities following mock treatment, suggesting that the depletion of these genes did not interfere with DNA

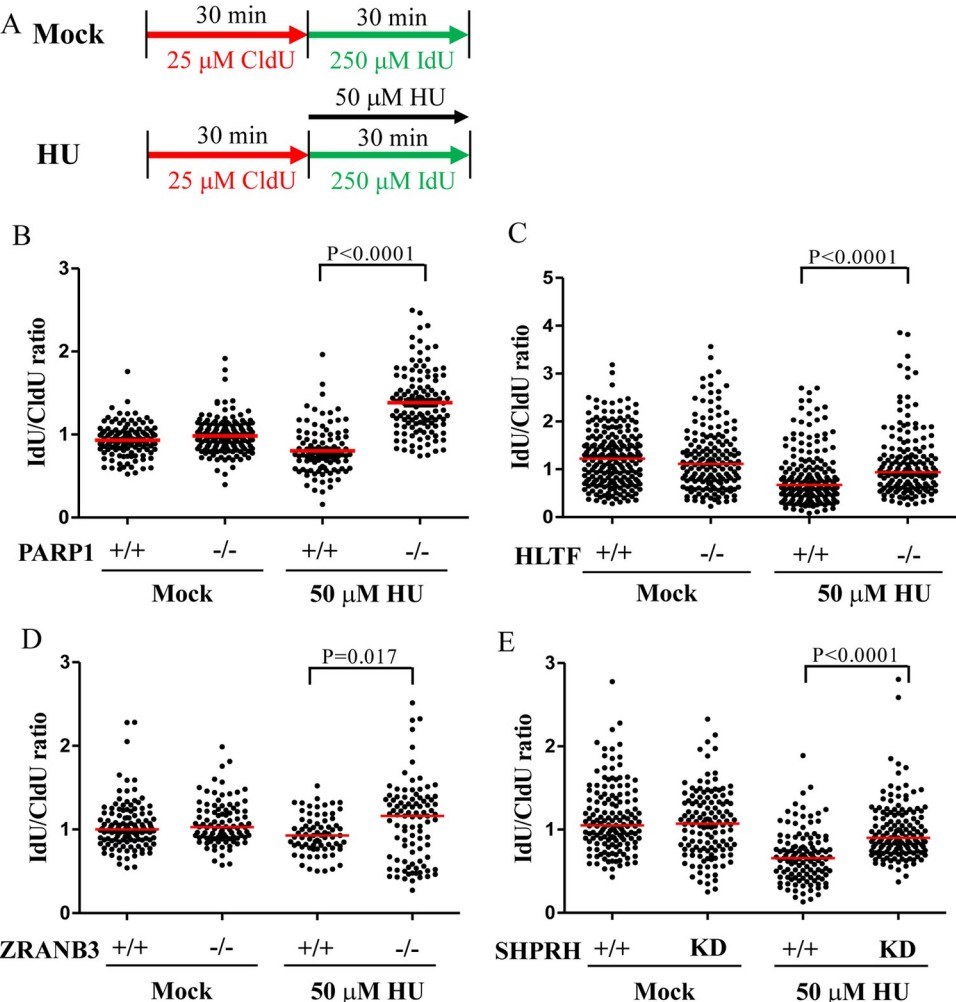

**Fig 5. PARP1 and DNA translocases restrain DNA replication upon HU-induced replication stress.** (A) Labeling protocols for DNA fiber analysis. (B)-(E) Quantitation of IdU/CldU track length ratios derived from each cell line. At least 100 DNA fibers derived from each cell line were measured. (Raw DNA fiber data in S5 Data).

replication progression (S15B, S15E, S16A, S16D, S16G and S16J Figs). However, the KO and KD cell lines all showed stronger Cy5 intensity than wild-type T24 cells upon HU treatment (S15C, S15F, S16B, S16E, S16H and S16K Figs), suggesting that PARP1, HLTF, SHPRH, and ZRANB3 restrain DNA replication progression upon replication stress, which is consistent with DNA fiber analysis.

## The levels of DNA damage severity affect the outcome of fork progression

HU and MMS represent two different types of replication stress. During a 30-min treatment, HU limits DNA replication by depleting cellular dNTP without inducing DNA lesions [52], whereas MMS blocks DNA replication by inducing DNA lesions without affecting cellular dNTPs levels [4]. Previous studies demonstrate that MMS treatment further slows fork progression in HLTF-depleted cells compared with wild-type cells [31,48], and the cellular response is speculated to differ according to the type of replication stress. To reconcile these results, we considered the possibility that differences in replication stress severity rather than

replication stress type leads to different outcomes. Treatment with 50 μM HU-induced relatively mild replication stress compared with 0.01% MMS. Treatment with 50 μM HU only barely or marginally induced the phosphorylation of CHK1, CHK2, and H2AX (Fig 6A). By contrast, treatment with 0.01% MMS greatly induced CHK1, CHK2, and H2AX phosphorylation (Fig 6A). Using the same 0.01% MMS treatment (Fig 7A), the replication track length was further reduced in the PARP1-KO, HLTF-KO, ZRANB3-KO, and SHPRH-KD cells compared with treatment with 50 μM HU (Fig 7B–7E). These results were also verified by using Cy5-labeled replication tracks (S15D, S15G, S16C, S16F, S16I and S16L Figs). We observed similar results in the U2OS cell line, suggesting that slowed fork progression induced by 0.01% MMS treatment was common across cell lines (S17 Fig).

To test whether lower-dose MMS treatment resulted in different outcomes, we treated cells with 0.001% MMS which only marginally induced CHK1, CHK2, and H2AX phosphorylation, similar to the outcome observed for 50 μM HU treatment (Fig 6A). Strikingly, PARP1-KO, HLTF-KO, and SHPRH-KD cells showed longer replication tracks upon 0.001% MMS treatment compared with control cells (Fig 7F). Therefore, our data resolved a discrepancy with the reports from past studies. We conclude that the severity of DNA damage affects the outcomes of fork progression. Combined with previous studies, these data indicate that the restraint of DNA replication by PARP1 and DNA translocases under replication stress conditions induced by HU, camptothecin, or MMS treatment represents a common phenomenon shared across different types of replication stress.

## PARP1-KO, HLTF-KO, and SHPRH-depleted cells have high levels of CHK1/CHK2 phosphorylation upon 0.01% MMS treatment

PARP1-KO, HLTF-KO, and SHPRH-KD cells slowed fork progression upon 0.01% MMS treatment. We tested whether this decrease in fork progression was due to the induction of higher levels of CHK1 and CHK2 phosphorylation. Cells were treated with 0.01% MMS for 30 min and the levels of phospho-CHK1 and phospho-CHK2 were determined by confocal microscopy. We observed similarly low levels of CHK1/CHK2 phosphorylation among wild type, PARP1-KO, HLTF-KO, and SHPRH-depleted cells following the mock treatment (Figs 6B, 6C, S18A and S18B), whereas 0.01% MMS treatment significantly increased CHK1/CHK2 phosphorylation in PARP1-KO, HLTF-KO, and SHPRH-depleted cells compared with wild type cells (Figs 6B, 6C, S18A and S18B). Our results suggest that higher levels of CHK1/CHK2 phosphorylation in PARP1-KO, HLTF-KO, and SHPRH-depleted cells could trigger a stronger intra-S phase checkpoint response, leading to a further decrease in DNA replication. These data also suggest that the mild replication stress induced by 50 μM HU or 0.001% MMS restrains fork progression without activating checkpoint signaling.

## PARP1- and DNA translocase-depleted cells are defective in DNA repair, resulting in higher levels of DSBs

To test whether PARP1-KO, HLTF-KO, and SHPRH-depleted cells present with defective DNA repair mechanisms, we performed a DNA recovery assay. We pulse-treated cells with 0.01% MMS for 1 hour. After MMS removal, the cells were allowed to recover for 6 or 24 hours (Fig 8A) and harvested for western blotting. As shown in Fig 8B–8D, the PARP1-KO, HLTF-KO, and ZRANB3-KO cells showed higher levels of CHK1 and CHK2 phosphorylation and γH2AX in the immunoblotting analysis. We further generated an HLTF/SHPRH double-depletion cell line, which showed higher levels of γH2AX and CHK1/CHK2 phosphorylation than wild-type cells (Fig 8E).

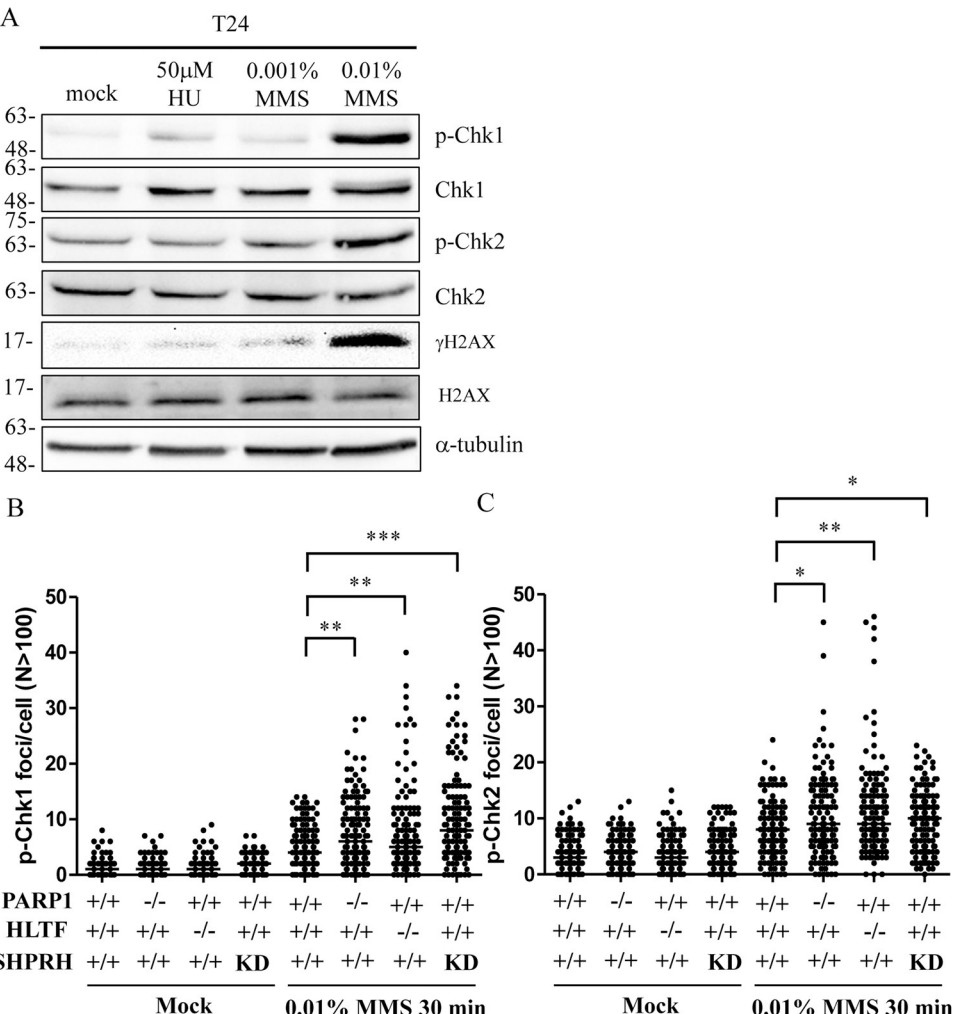

**Fig 6. Higher levels of CHK1 and CHK2 phosphorylation are induced by 0.01% MMS than 50 μM HU or 0.001% MMS.** (A) Cells were treated with 50 μM HU, 0.001% MMS, or 0.01% MMS for 1 hour. Cells were fixed and immunostained with antibodies as indicated. (B)(C) PARP1-, HLTF, and SHPRH-deficient cells showed higher levels of phospho-CHK1 and phospho-CHK2 than control cells. (Raw p-CHK1/pCHK2 data in S6 Data).

To determine whether HLTF and SHPRH act synergistically, we compared the levels of DSBs between HLTF/SHPRH single- and double-depleted cells. We tested the levels of 53BP1 foci using confocal microscopy. Cells were treated with 0.01% MMS for 1 hour to induce DNA lesions. After the removal of MMS, the cells were allowed to recover for 24 hours. The HLTF/SHPRH double-depleted cells exhibited a higher intensity of γH2AX and 53BP1 (S19A and S19B Fig) and an increase in the number of 53BP1 foci compared with wild-type cells (S19C Fig). The γH2AX and 53BP1 foci were highly correlated (S20 Fig). Our results suggest that HLTF and SHPRH act synergistically to recover from MMS treatment.

## Discussion

In this study, we identified a novel interaction between PARP1 and DNA translocases, with HLTF showing the strongest interaction with PARP1 among DNA translocases. The interaction not only recruits DNA translocases to DNA damage sites to restrain DNA replication but

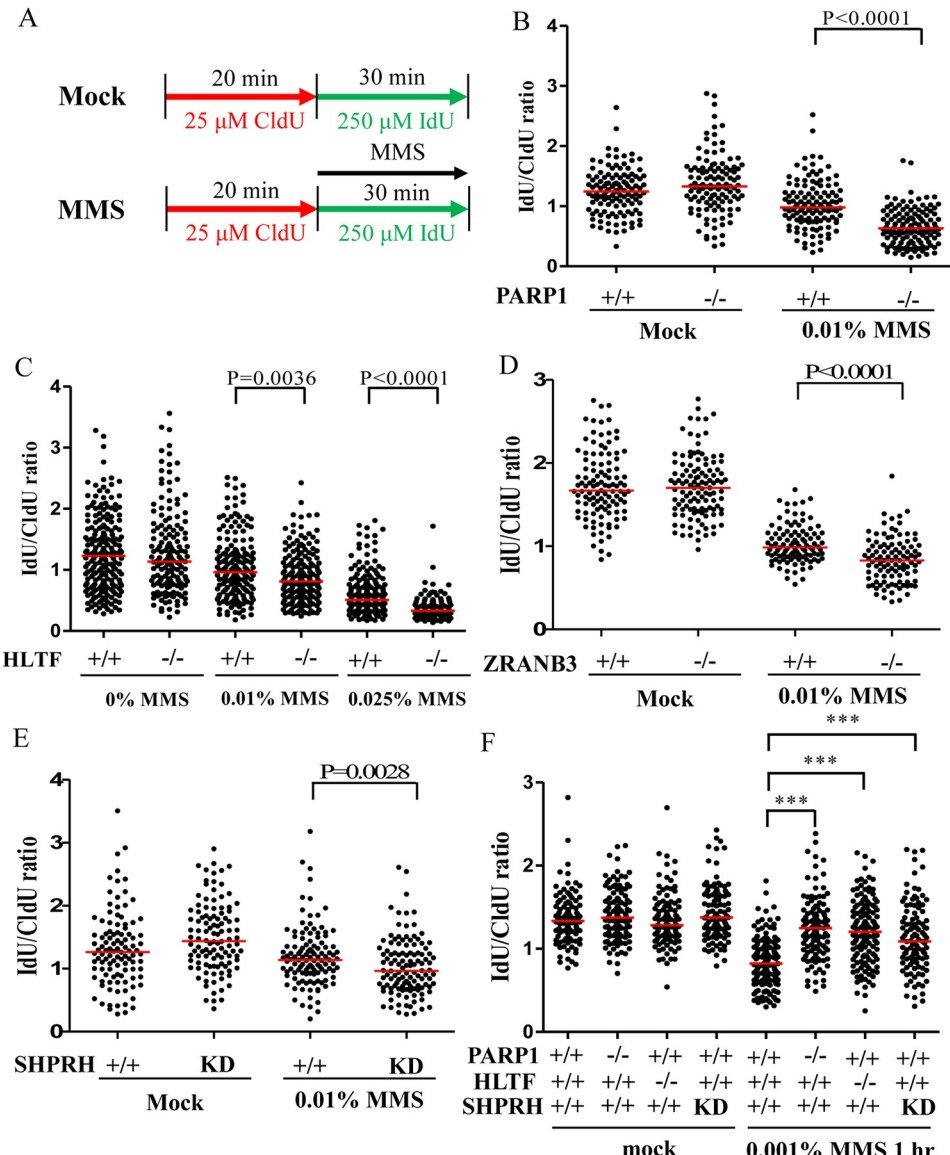

**Fig 7. DNA fiber analysis upon MMS-induced replication stress.** (A) Labeling protocols for DNA fiber analysis. (B)-(F) Quantitation of IdU/CldU track length ratios derived from each cell line. At least 100 DNA fibers derived from each cell line were measured. (Raw DNA fiber data in S7 Data).

also facilitates DNA repair. In support of this notion, we found that the PARP1- and DNA translocase-deficient cells displayed significantly reduced DNA repair efficiency. After 24 hours recovery from MMS-induced DNA damage, increased levels of γH2AX, 53BP1 foci, and CHK1 and CHK2 phosphorylation were detected in gene-depleted cells than in wild-type cells. We believed that these DNA translocases convert stalled forks into reversed forks to protect stalled fork from collapse and facilitate DNA repair.

Previous studies suggest that HLTF and SHPRH are recruited to replication forks through interactions with PCNA [26,29]. Here, we further revealed that PARP1 interacts with DNA translocases, including HLTF, SHPRH, ZRANB3, and SMARCAL1, with HLTF showing the strongest interaction with PARP1. Consistent with coimmunoprecipitation assay, PLA assay

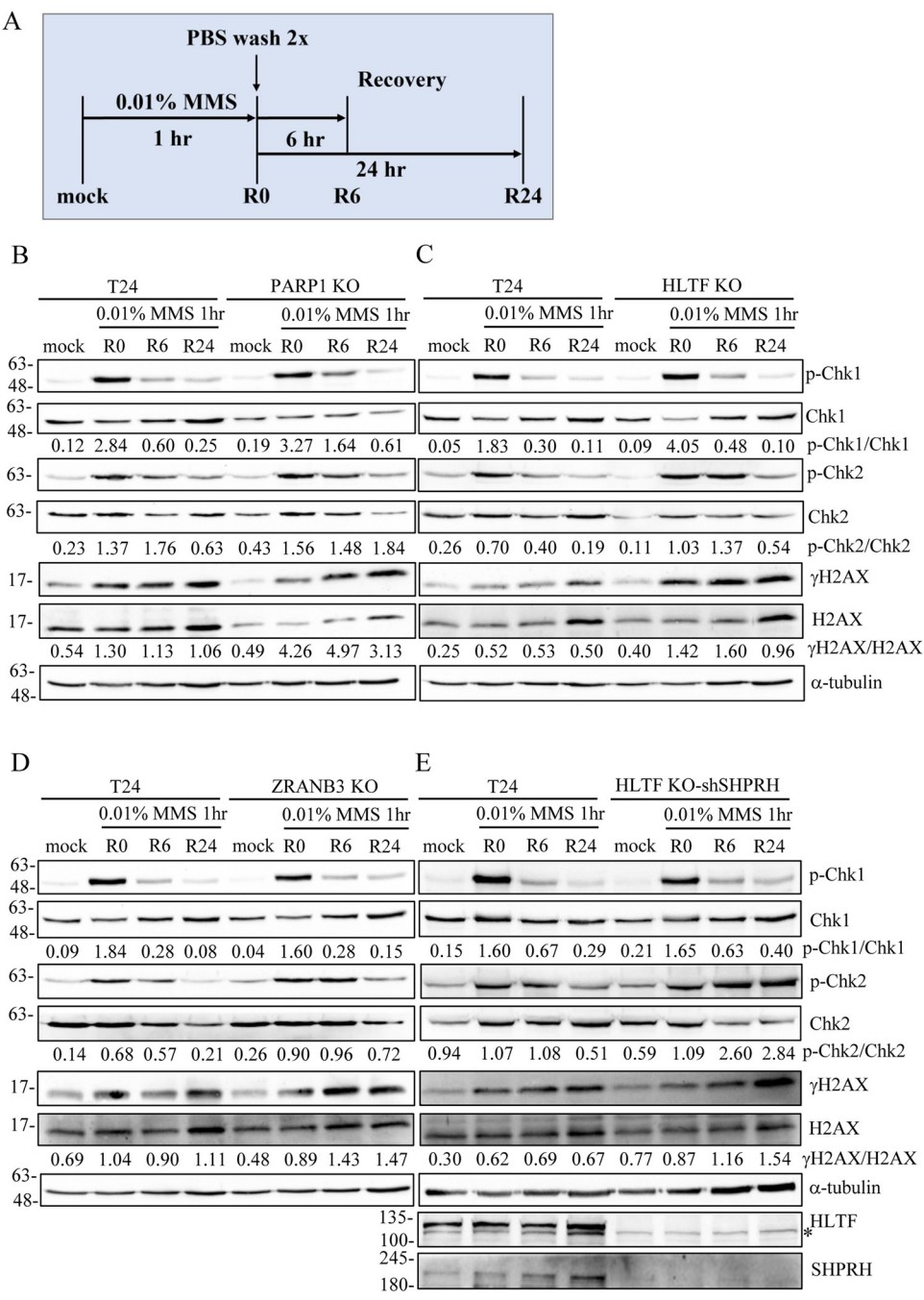

**Fig 8. PARP1-, HLTF-, and ZRANB3-depleted cells show higher levels of γH2AX and checkpoint activation than control cells after MMS treatment.** (A) The protocol of DNA damage recovery assay. Cells were treated with 0.01% MMS for 1 hour (R0), followed by recovery in fresh medium for 6 (R6) and 24 (R24) hours. Cells were fixed and immunostained with antibodies as indicated. (B)-(E) Immunostaining of DNA damage recovery assay. Proteins were detected with specific antibodies as indicated. Asterisk (*) indicates non-specific band.

also revealed a similar trend with PARP1/HLTF showing the most PLA foci among the DNA translocases. Since DNase I or EtBr treatment did not disrupt these interactions, it suggests these interactions are not mediated through DNA. Interestingly, MMS or UV treatment increases PARP1/DNA translocase PLA foci. Using SIRF assay, we also found that MMS or

UV treatment induces an increase of HLTF, SHPRH, ZRANB3, and SMARCAL1 levels at replication forks. We think since PARP1/DNA translocases complex is enriched at damaged forks upon MMS or UV treatment, the numbers of HLTF/PARP1, SHPRH/PARP1 and ZRANB3/PARP1 PLA foci increase after MMS or UV treatment. Consistent with our results, a previous study also revealed that UV irradiation reduces fork progression and increases frequency of reversed forks using electron microscopy [25]. Taken together, these results suggest that replication slowing and the formation of reversed forks are global responses to genotoxic treatments and this process requires PARP1/DNA translocases recruitment to damaged forks.

The depletion of PARP1 significantly reduced the numbers of these DNA translocase/biotin PLA foci (Fig 1A–1H), whereas the introduction of exogenous PARP1 to PARP1-KO cells was able to rescue these DNA translocase/biotin PLA foci (Fig 2A–2D). Interestingly, fewer HLTF/biotin, SHPRH/biotin, ZRANB3/biotin, and SMARCAL1/biotin PLA foci were found in nucleus than PCNA/biotin PLA foci (Figs 1A–1H and S3), suggesting that these DNA translocases are not components of every replisome; instead, these DNA translocases may be recruited specifically to damaged replication forks, resulted from intracellular ROS or exogenous genotoxic insults. The introduction of a PARP1-K893I mutant with disrupted PAR activity failed to fully restore the recruitment of DNA translocases in PARP1-KO cells. The inhibition of PARP1 with olaparib also significantly reduced HLTF/biotin and SHPRH/biotin PLA foci. These data indicate that PAR activity is critical for the recruitment of DNA translocases to damaged forks. However, an analysis of the HLTF and SHPRH protein sequences did not reveal any PAR-binding motifs, macrodomains, PAR-binding zinc fingers (PBZ), WWE domains, or BRCT domains which are able to interact with PAR [53]. Therefore, PARP1 and DNA translocases could form a complex and the interaction is not mediated through PAR modification. However, PARP1 activity is critical for the recruitment of the PARP1/DNA translocases complex to damaged forks (Figs 2 and S4 and S6–S10). We speculate that PARP1 activity regulates chromatin compaction by recruiting ALC1 and APLF [54,55] or by PARylating histones [56–58], which results in local chromatin structure change and allows the access of DNA translocases to chromatin. Consistent with this notion, HLTF is found in APLF and ALC1 complex in a large scale affinity purification mass spectrometry project [59]. Both HLTF and SHPRH interact with PCNA [26]. The HIRAN domain of HLTF interacts with 3' ssDNA [35]. These multivalent interactions induce the recruitment of PARP1/DNA translocases complex to damaged forks. However, the detailed mechanism of how PARP1 activity coordinates the recruitment of DNA translocases to damage forks requires further investigation in the future. Since PARP1 contains a DNA binding domain (DBD) that binds and is activated by stalled forks containing small gaps [16], we speculate that in addition to its role in BER, PARP1 could serve as a replication stress sensor that binds to the stressed replication fork and recruits DNA translocases to the stressed fork to restrain fork progression upon replication stress.

HLTF and SHPRH share structural and functional similarity [28]. Both of them are ubiquitin E3 ligases that polyubiquitinate PCNA. To date, only HLTF has been characterized to be an ATP-dependent DNA translocase that catalyze fork reversal formation. It remains unclear whether SHPRH has such function. Our DNA fiber analysis, together with results derived from EdU-conjugated Cy5 fluorescence microscopy revealed that SHPRH restrains fork progression upon replication stress, a function similar to HLTF. Interestingly, an in silico analysis also revealed that N-terminal domain of SHPRH also includes a potential HIRAN-like domain, which could play a role in recognition of 3'DNA ends [60]. Therefore, SHPRH could contain DNA translocase activity. Alternatively, SHPRH could serve as a helper or scaffold protein to facilitate fork reversal. It requires further biochemical study to verify its function.

Replication stress results in DNA translocases-mediated DNA replication restraint. Here, we revealed that the restraint of replication fork progression is a common mechanism in response to different DNA damaging-inducing agents. Previous studies showed that HLTF-depletion slowed fork progression upon MMS treatment due to high levels of DNA damage [31,48], which activates intra-S-phase, leading to further replication fork progression restraint. By contrast, low levels of replication stress (e.g. 50 μM HU or 0.001% MMS) restrains fork progression through the recruitment of DNA translocases without activating the intra-S phase checkpoint pathway. These findings suggest that the recruitment of DNA translocases to stalled replication forks does not require checkpoint activation but depends on the binding of PARP1 to stalled forks. However, we cannot exclude the possibility that even marginal or local CHK1 activation is able to promote fork reversal.

We found that PARP1- and DNA translocase-depleted cells display DNA damage recovery defects, suggesting that DNA translocases-mediated replication fork reversal protects stalled replication forks from collapse and facilitates DNA repair by BER. In conclusion, we uncovered a DNA damage response mechanism in which PARP1 binds to stalled replication forks and recruits DNA translocases. HLTF and SHPRH mediated PCNA polyubiquitination, which could facilitate ZRANB3 recruitment to damaged forks. These reversed forks not only protect damaged forks from collapse, but also facilitate DNA repair.

## Material and methods

### Cell culture

The bladder cancer cell line T24 (ATCC HTB) was maintained in McCoy's 5A media supplemented with 10% fetal bovine serum (CORNING), 1% glutamine (CORNING), and 1% penicillin/streptomycin (CORNING) in a 37°C incubator containing 5% CO2. The human embryonic kidney cell line HEK293T was maintained in DMEM medium supplemented with 10% fetal bovine serum, 1% glutamine, and 1% penicillin/streptomycin in a 37°C incubator containing 5% CO2.

### SIRF assay

Cells were harvested in 8-well chamber slides (Millipore) at a density of $3 \times 10^4$ cells and pulse-labeled with 10 μM EdU for 15 minutes followed by treatment with 1.2 mM (0.01%) MMS for 1hour or treatment with 60 J/m$^2$ UV irradiation (254 nm, Ultraviolet crosslinker CL-1000, UVP Inc). we then used extraction buffer (25 mM HEPES, pH7.4, 50 mM NaCl, 3 mM MgCl2,1 mM EDTA, 0.3 M sucrose, 0.5% Triton X-100) to remove the cytoplasmic fraction. The cells were fixed with 3.5% paraformaldehyde in PBS for 30 minutes at room temperature. The click reaction was performed with biotin-azide for 30 minutes and quenched with blocking solution for 30 minutes. Two primary antibodies against biotin and target proteins were used to detect nascent synthesized DNA and the target proteins, respectively. The primary antibodies used in this study are listed in S2 Table. Two oligonucleotide-conjugated secondary antibodies, anti-rabbit PLUS and anti-mouse MINUS antibodies (Duolink, Sigma-Aldrich), were incubated with samples for 60 minutes at 37°C. Ligases and DNA polymerases were then applied to the samples according to the manufacturer's protocol (Duolink, Sigma-Aldrich). Finally, the samples were mounted with Duolink In Situ Mounting Medium with DAPI for 15 minutes at room temperature and analyzed using a Nikon Eclipse 80i microscope equipped with a Plan Fluor 40x/0.75 DIC M/N2 objective. The resulting images were measured using NIS Elements D4.20.00 software (Nikon).

## Immunofluorescence microscopy

Cells were cultured in 8-well chamber slides at a density of 3 x 10^4 cells/well overnight and treated with 0.01% MMS for 1 hr. Then, the cells were fixed with 3.5% paraformaldehyde for 30 minutes at room temperature. The fixed cells were incubated in blocking buffer (10% FBS, 0.1% Triton X100 in PBS) for 30 minutes at room temperature and further incubated with anti-γH2AX antibody (1:100, Millipore, 05–636) and anti-53BP1 antibody (1:100, Abcam, ab36823) at 4°C overnight. Alexa Fluor 594-labeled goat anti-mouse IgG (H+L) antibody (1:500, ThermoFisher Scientific, A-11032) and Alexa Fluor 488-labeled goat anti-rabbit IgG (H+L) antibody (1,500, ThermoFisher Scientific, A-11001) were then used to detect γH2AX and 53BP1 foci. ProLong Gold Antifade Mountant with DAPI (ThermoFisher Scientific, P36935) was used to stain nuclei. Images were captured using a Zeiss LSM 780 confocal Microscope, and the intensity of γH2AX and 53BP1 was quantified with ZEN 3.3 (blue edition) in at least 100 cells were measured per cell line.

## DNA fiber analysis

Cells were cultured in a 100-mm culture dish at a density of 2 x 10^6 cells for 16 hours and pulse-labeled with 25 μM CldU for 20 minutes, followed by treatment with MMS or HU and pulse-labeling with 250 μM IdU for 30 minutes. The nuclear fraction was isolated with buffer A (10 mM HEPES, pH 7.9, 10 mM KCl, 1.5 mM MgCl2, 0.34 M sucrose, 10% glycerol). The nuclear fraction was plated onto glass slides and lysed with spreading buffer. Each slide was tilted 15° to 30° to allow the droplet to slowly run down the slide, spreading DNA fibers along the slide. The DNA fibers were fixed in a 3:1 methanol/acetic acid buffer for 10 minutes, denatured with 2N HCl for 1 hour at room temperature, and blocked in PBS buffer (1% BSA, 0.1% Tween20 in PBS). Slides were incubated at 4°C overnight with rat anti-BrdU primary antibody (1:200, Abcam, ab6353) against CldU and the mouse anti-BrdU primary antibody (1:200, BD Biosciences, 347580) against IdU. The goat anti-rat Alexa Fluor-594-conjugated secondary antibody (1:500, ThermoFisher Scientific, 1301853) and the anti-mouse Alexa Fluor-488-conjugated secondary antibody (1,500, ThermoFisher Scientific, 1613346) were added to the slides for 90 minutes at room temperature. Images were acquired by a Nikon Eclipse 80i microscope equipped with a Nikon Plan Apo 100x/1.40 Oil DIC objective. At least 100 fibers were analyzed in each cell line using NIS Elements D4.20.00 software (Nikon), and graphs were plotted with GraphPad Prism software (Version 5.0).

The detailed Materials and Methods are described in S1 Text.

## Supporting information

**S1 Text. Supplementary Materials and Methods.**
(PDF)

**S1 Fig. The immunoblotting of the knockdown or knockout cell lines.** (A)-(E) The knockdown (KD) or knockout (KO) of each gene was verified by western blot analysis using specific antibodies as indicated. Two shRNAs, KD#1 and KD#2, against SMARCAL1 were used as indicated.
(TIF)

**S2 Fig. Sister chromatid exchange (SCE) in wild-type and PARP1-KO T24 cells.** (A) SCE analysis of control T24 cells and PARP1-KO T24 cells. SCE was scored in 50 metaphase cells in each cell line. (B) Representative images of SCE derived from each cell line. (C) Overexpression of DNA binding domain of PARP1 increases SCE frequency. HONE6 cells were transfected with an empty vector pEGFP or pEGFP-DBD. The expression of DBD-GFP fusion

protein was determined using western blotting with an anti-GFP antibody. (D) Representative images of SCE for each cell line. (E) Quantification of SCE was scored from at least 50 metaphase cells per cell line. (Raw SCE data in S8 Data).
(TIF)

**S3 Fig. PCNA SIRF assay.** (A) Representative images of control PLA and PCNA SIRF assays. PLA foci are shown in red, and DAPI staining is shown in blue. (B) Distribution of PLA foci from each cell derived from (A). At least 100 cells from each condition were measured. (C) The numbers of PLA foci from each cell were classified into four groups: 0, 1–50, 51–100, and >100 foci. The distributions of each group are indicated in the plot. (Raw SIRF data in S9 Data).
(TIF)

**S4 Fig. The levels of DNA translocases are reduced at damaged replication forks in PARP1-KO cells.** (A)-(D) The numbers of PLA foci (Fig 1A–1D) were classified into four groups: 0, 1–5, 6–10, and >11 foci, and the distributions of each group are shown in the plots. (Raw SIRF data in S1 Data).
(TIF)

**S5 Fig. UV treatment induces an increase of DNA translocases at damaged forks.** (A) Representative images of PLA foci of each DNA translocase in mock or UV-treated T24 cells. Cells were treated with 60 J/cm$^2$ of UV irradiation. The association of each DNA translocase with replication forks was determined by the SIRF assay. (B) Distribution of PLA foci derived from a, respectively. At least 200 cells from each condition were measured. (C) The number of PLA foci was classified into four groups: 0, 1–5, 6–10, and >11 foci, and the distributions of each group were shown in the plot. (Raw SIRF data in S10 Data).
(TIF)

**S6 Fig. The levels of DNA translocases are reduced at damaged replication forks in PARP1 knockdown T24 cells.** (A)(D) Representative images of HLTF and SHPRH PLA foci in wild-type and PARP1-knockdown T24 cells. The expression of PARP1 was depleted using shRNA lentivirus. Cells were treated with 0.01% MMS for 1 hour. The association of each protein with replication forks was determined by the SIRF assay. (B)(E) Distribution of HLTF and SHPRH PLA foci derived from (A)(D), respectively. At least 200 cells from each condition were measured. (C)(F) The numbers of PLA foci were classified into four groups: 0, 1–5, 6–10, and >11 foci, and the distributions of each group are shown in the plot. (Raw SIRF data in S11 Data).
(TIF)

**S7 Fig. The levels of DNA translocases are reduced at damaged replication forks in PARP1 knockdown T24 cells.** (A)(D) Representative images of ZRANB3 and SMARCAL1 PLA foci in wild-type and PARP1-knockdown T24 cells. The expression of PARP1 was depleted using shRNA lentivirus. Cells were treated with 0.01% MMS for 1 hour. The association of each protein with replication forks was determined by the SIRF assay. (B)(E) Distribution of ZRANB3 and SMARCAL1 PLA foci derived from (A)(D), respectively. At least 200 cells from each condition were measured. (C)(F) The numbers of PLA foci were classified into four groups: 0, 1–5, 6–10, and >11 foci, and the distributions of each group are shown in the plot. (Raw SIRF data in S11 Data).
(TIF)

**S8 Fig. DNA translocase levels at damaged replication forks are rescued by the introduction of PARP1 in PARP1-KO cells.** (A)-(D) Representative images of HLTF, SHPRH, ZRANB3, and SMARCAL1 PLA foci, respectively, in PARP1-KO, PAPR1-rescue, and

PARP1-K893I expressing T24 cells. The pLNCX vectors carrying wild-type PARP1 or PARP1-K893I mutant, respectively, were packaged into retrovirus particles in GP2-293 cell line. PARP1-KO T24 cells were infected with these retroviruses to stably express wild-type PARP1 or PARP1-K893I mutant. Retrovirus carrying the empty vector (vec) was used as the control. Cells were treated with 0.01% MMS for 1 hour. The association of each protein with replication forks was determined by the SIRF assay. (Raw SIRF data in S2 Data). (TIF)

**S9 Fig. DNA translocase levels at damaged replication forks are rescued by the introduction of PARP1 in PARP1-KO cells.** (A)-(D) The numbers of PLA foci were classified into four groups: 0, 1–5, 6–10, and >11 foci, and the distributions of each group are shown in the plot. (Raw SIRF data in S2 Data). (TIF)

**S10 Fig. HLTF and SHPRH levels are reduced at damaged forks in olaparib- treated T24 cells.** (A)(D) Representative images of HLTF and SHPRH PLA foci in mock or olaparib-treated T24 cells. Cells were treated with olaparib for 2 hours, followed by 0.01% MMS treatment for 1 hr. The association of each protein with replication forks was determined by the SIRF assay. (B)(E) Distributions of HLTF and SHPRH-PLA foci derived from a, d, respectively. At least 200 cells from each condition were measured. (C)(F) The numbers of PLA foci were classified into four groups: 0, 1–5, 6–10, and >11 foci, and the distributions of each group are shown in the plot. (Raw SIRF data in S12 Data). (TIF)

**S11 Fig. The schematic representation of SHPRH constructs.** The helicase ATP binding domain first part (HAB1), H15, PHD, helicase ATP binding domain second part (HAB2), RING, and helicase C-terminal domain (HCT) are based on UniProt Knowledgebase (UniProtKB) analysis. (TIF)

**S12 Fig. PARP1 domains interact with DNA translocases.** (A) The schematic representation of PARP1 constructs. The DNA binding domain (DBD), BRCT domain, and catalytic domain (CAT) are based on UniProt Knowledgebase (UniProtKB) analysis. (B) HEK293T cells were transfected with various GFP-PARP1 constructs. The GFP-PARP1 fusion proteins were immunoprecipitated with a GFP antibody followed by protein G-agarose pulldown. The immunoprecipitates were then subjected to immunoblotting analysis with specific antibodies as indicated. Input represents 5% of total cell lysates. (TIF)

**S13 Fig. PARP1 interacts with DNA translocases *in vivo*.** The number of PLA foci (Fig 3C) was classified into four groups: 0, 1–5, 6–10, and >11 foci, and the distributions of each group are shown in the plot. (Raw PLA data in S3 Data). (TIF)

**S14 Fig. PARP1 interacts with DNA translocases *in vivo*.** (A) Representative images of PARP1/translocases PLA foci in mock or UV-treated T24 cells. T24 cells were treated with 60 J/cm$^2$ of UV irradiation. (B) Distributions of PARP1/HLTF, PARP1/SHPRH, PARP1/ZRANB3, and PARP1/SMARCAL1 PLA foci derived from a. At least 200 cells from each condition were measured. (C) The numbers of PLA foci were classified into four groups: 0, 1–5, 6–10, and >11 foci, and the distributions of each group are shown in the plot. (Raw PLA data in S13 Data). (TIF)

**S15 Fig. 0.01% MMS- and 50 μM HU-induced replication stress have different effect on the progression of replication tracks.** (A) The schematic representation of EdU click reaction assay. (B)(C)(D) Representative images of Cy5 fluorescent intensity derived from each cell line and treatment. Wild-type and PARP1-KO T24 cells were labeled with EdU for 15 min (mock), followed by treatment with 50 μM HU or 0.01% MMS for 1 hour before fixation. Cy5 (pink) was conjugated to EdU by the click reaction, and images were acquired using a Zeiss LSM 780 confocal microscope. (E)(F)(G) The intensity of Cy5 was quantified from at least 100 cells per condition. (Raw Cy5 data in S14 Data).
(TIF)

**S16 Fig. 0.01% MMS- and 50 μM HU-induced replication stress have different effect on the progression of replication tracks.** (A)(B)(C)(G)(H)(I) Representative images of Cy5 fluorescent intensity derived from each cell line and treatment. The wild type, HLTF-KO, ZRANB3-KO, and SHPRH-knockdown (shSHPRH) T24 cells were labeled with EdU for 15 min (mock), followed by treatment with 50 μM HU or 0.01% MMS for 1 hour before fixation. Cy5 (pink) was conjugated to EdU by the click reaction, and images were acquired using a Zeiss LSM 780 confocal microscope. (D)(E)(F)(J)(K)(L) The intensity of Cy5 was quantified from at least 100 cells per condition. (Raw Cy5 data in S14 Data).
(TIF)

**S17 Fig. Depletion of PARP1, HLTF, or SHPRH further reduced replication tracks following 0.01% MMS treatment in U2OS cells.** Quantitation of IdU/CldU track length ratios derived from each cell line. At least 100 DNA fibers derived from each cell line were measured. (Raw DNA fiber data in S15 Data).
(TIF)

**S18 Fig. 0.01% MMS induces high levels of CHK1 and CHK2 phosphorylation in PARP1-KO, HLTF-KO, and SHPRH-depleted cells compared with control cells.** (A)(B) The numbers of phospho-CHK1 and phospho-CHK2 foci from each cell were classified into four groups: 0, 1–5, 6–10, >11 foci. The distributions of each group are indicated in the plot. (Raw pCHK1/pCHK2 data in S6 Data).
(TIF)

**S19 Fig. The HLTF-KO and HLTF/SHPRH-double depleted cells show higher levels of DSBs.** (A)(B) The intensity of 53BP1 and γH2AX in each cell was quantified using ZEN 3.3 software. Cells were treated with 0.01% MMS for 1 hour, followed by recovery in fresh medium for 24 hours. Cells were fixed and immunostained with 53BP1 and γH2AX antibodies. At least 100 cells from each cell line were quantified. (C) The number of 53BP1 foci from each cell was classified into two groups: <10 and >10, and the distributions of each group are indicated in the plot. (Raw 53BP1/γH2AX data in S16 Data).
(TIF)

**S20 Fig. The γH2AX and 53BP1 foci are highly correlated.** The confocal microscopy results of 53BP1 and γH2AX derived from each cell line. Cells were chronically treated with 0.01% MMS for 1 hr, followed by recovery in fresh medium for 24 hr. Cells were fixed and immunostained with 53BP1 and γH2AX antibodies.
(TIF)

**S1 Table. The list of targeting sequences of shRNA used in this study.**
(PDF)

**S2 Table. The list of antibodies used in this study.**
(PDF)

**S1 Data. Raw SIRF data.**
(XLSX)

**S2 Data. Raw SIRF data.**
(XLSX)

**S3 Data. Raw PLA data.**
(XLSX)

**S4 Data. Raw SIRF data.**
(XLSX)

**S5 Data. Raw DNA fiber data.**
(XLSX)

**S6 Data. Raw pCHK1/pCHK2 data.**
(XLSX)

**S7 Data. Raw DNA fiber data.**
(XLSX)

**S8 Data. Raw SCE data.**
(XLSX)

**S9 Data. Raw SIRF data.**
(XLSX)

**S10 Data. Raw SIRF data.**
(XLSX)

**S11 Data. Raw SIRF data.**
(XLSX)

**S12 Data. Raw SIRF data.**
(XLSX)

**S13 Data. Raw PLA data.**
(XLSX)

**S14 Data. Raw Cy5 data.**
(XLSX)

**S15 Data. Raw DNA fiber data.**
(XLSX)

**S16 Data. Raw 53BP1/γH2AX data.**
(XLSX)

## Acknowledgments

We thank Hung-Wen Li, Nei-Li Chan, and all members of our laboratories for their critical reading of the manuscript prior to publication. We kindly thank Wen-Pin Su, Ming-Daw Tsai, and Shun-Fen Tzeng for their support.

## Author Contributions

**Conceptualization:** Hungjiun Liaw.

**Investigation:** Yen-Chih Ho, Chen-Syun Ku, Siang-Sheng Tsai, Jia-Lin Shiu, Yi-Zhen Jiang, Hui Emmanuela Miriam, Han-Wen Zhang, Yen-Tzu Chen, Hungjiun Liaw.

**Methodology:** Yen-Chih Ho, Wen-Tai Chiu, Song-Bin Chang, Che-Hung Shen, Peter Chi.

**Resources:** Kyungjae Myung, Peter Chi.

**Supervision:** Hungjiun Liaw.

**Writing – original draft:** Hungjiun Liaw.

**Writing – review & editing:** Hungjiun Liaw.

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
