## [Decision Letter · Decision Letter 0]

20 Aug 2022

Dear Dr Liaw,

Thank you very much for submitting your Research Article entitled 'PARP1 recruits DNA translocases to restrain DNA replication and facilitate DNA repair' to PLOS Genetics.

The manuscript was fully evaluated at the editorial level and by independent peer reviewers. The reviewers appreciated the attention to an important problem, but raised some substantial concerns about the current manuscript. Based on the reviews, we will not be able to accept this version of the manuscript, but we would be willing to review a much-revised version. We cannot, of course, promise publication at that time.

If you decide to revise the manuscript for further consideration at PLOS Genetics, please aim to resubmit within the next 60 days, unless it will take extra time to address the concerns of the reviewers, in which case we would appreciate an expected resubmission date by email to plosgenetics@plos.org.

[LINK]

We are sorry that we cannot be more positive about your manuscript at this stage. Please do not hesitate to contact us if you have any concerns or questions.

Yours sincerely,

Shan Zha

Guest Editor

PLOS Genetics

Peter McKinnon

Section Editor

PLOS Genetics

Reviewer's Responses to Questions

**Comments to the Authors:**

Reviewer #1: Evaluation of manuscript PGENETICS-D-22-00803, entitled: “PARP1 recruits DNA translocases to restrain DNA replication and facilitate DNA repair” by Yen-Chih Ho et al.,

Synopsis and Significance.

The manuscript concerns the complex process of replication fork reversal, which is critical for genome stability as it restrains progression of DNA replication to rescue arrested DNA synthesis and prevent fork collapse and facilitate DNA repair. In this process PARP1 plays an important but still only partly clarified role. The authors have addressed the interaction of PARP1 with (presumed) DNA translocases, including HLTF, SHPRH, ZRANB3, and SMARCAL1, focusing on SHPRH. They reveal that this protein acts as a DNA translocase by its ability to restrain DNA replication stress and that PARP1 serves to recruit DNA translocases to enable repair. This work helps to better understand how replication stress is handled by the cellular DNA damage response machinery, and hence is relevant for genome stability and cancer. The work presented in the manuscript is of high quality (including technically delicate assays such as the SIRF methodology) and constitutes a substantial amount of experimental data. The findings are novel and appear solid.

Comments and questions:

1. Overall this referee finds the results indicating that PARP1 facilitates the recruitment of all 4 DNA translocases to damaged forks in a PAR-activity-dependent fashion shown in Fig 1, 2 and Fig. S1-6 are convincing and provided with all essential controls. Inhibiting PARP1 using Olaparib substantially reduced HLTF/biotin and SHPRH/biotin PLA foci, suggesting that PAR activity is critical for the recruitment of DNA translocases. However, besides inactivating PARP1, Olaparib also causes PARP1-DNA-bound complexes, which might interfere with replication complicating the interpretation.

2. The mapping of the interaction domains in SHPRH in Fig 3b and supplementary Fig. S8, shows that the N-terminal part of SHPRH (aa 1-606), containing SNF2 and H15 domains, interacts with PARP1, and exclude the C-terminal part (aa 1090-1683). However, the middle part (aa 606-1090) has not been tested. Hence, this does not rule out the possibility that this part containing the PHD and SNF2 domains does also bind PARP1 and could harbor important functions. Is there a reason why the authors have not examined this portion of the protein? An interesting additional question is where does SHPRH bind to PARP1 and do the other DNA translocases bind to the same or a different part of PARP1?

3. It is remarkable that MMS did not enhance the in vivo interactions between PARP1 and all DNA translocases: HLTF, SHPRH, ZRANB3 and SMARCAL1, whereas one would expect that the interaction after DNA damage would most likely be strengthened, when the translocases are implicated in the response to replication stress. This is even more striking considering that MMS treatment induced an increase in the numbers of HLTF/PARP1, SHPRH/PARP1 and ZRANB3/PARP1 PLA foci (fig. 3c,d and S9). What is the explanation for this observation? Is this also the case for other types of replication-blocking DNA lesions such as UV-induced cyclobutane pyrimidine dimers, which require translesion synthesis for bypass?

4. HLTF and SHPRH facilitate the loading of ZRANB3 to damaged replication forks in a PARP1-dependent fashion. Importantly, the authors demonstrate in two independent ways that SHPRH restrains DNA replication under conditions of HU-induced replication stress, providing the first evidence that this protein resembles its structurally-related cousin HLTF. Additionally, using different MMS concentration the investigators demonstrate that that HU, camptothecin, or MMS restrain DNA replication by the same mechanism, when similar DNA stress levels are induced, clarifying an apparent disparity in literature. The authors demonstrate that KO or depletion of PARP1, HLTF or SHPRH after MMS treatment activate Chk1 and Chk2 by phosphorylation (Fig. 6a) as well as γH2AX. Although overall the quality of immunoblots in the manuscript is high it appears that the γH2AX and the H2AX immunoblots are from different gels (See Fig. 6a). For proper comparison the same gel should be compared accompanied by the corresponding loading control (e.g., α-tubulin as used by the investigators), as done for p-Chk1 and Chk1 as well as pChk2 and Chk2. The same remark applies to Fig. 8b-d. Finally, the authors show that doubly depleted cells displayed elevated levels of 53BP1 foci (Fig. S14), suggesting that HLTF and SHPRH act synergistically to recover from MMS treatment.

5. In the Discussion the authors state that their findings “… explain the mechanism of PARP1 in fork progression restraint upon replication stress.” This is somewhat overstated: although the results are interesting and provide further insight into the mechanisms of replication restrain still many questions remain and further investigation is needed before it is fully explained.

Minor comment: The manuscript contains a number of grammatical errors.

In summary: The manuscript by Yen-Chih Ho et al., presents a substantial amount of work on interesting aspects of replication stress: the roles of PAPR1 and DNA translocases in replication restrain, which is very important for genome (in)stability. The results extend our knowledge about the involvement of various DNA translocases and of PARP1 in this process. Apart from the remarks made above, the experiments are generally of high quality and support most of the conclusions.

Reviewer #2: uploaded as attachment

**Have all data underlying the figures and results presented in the manuscript been provided?**

Reviewer #1: Yes

Reviewer #2: Yes

PLOS authors have the option to publish the peer review history of their article (what does this mean?). If published, this will include your full peer review and any attached files.

Reviewer #1: No

Reviewer #2: No

---

## [Decision Letter · Decision Letter 1]

26 Nov 2022

Dear Dr Liaw,

We are pleased to inform you that your manuscript entitled "PARP1 recruits DNA translocases to restrain DNA replication and facilitate DNA repair" has been editorially accepted for publication in PLOS Genetics. Congratulations! Before submit the final version, please address the minor changes on UVB source and technical deatils brought by the reviewers. 

Yours sincerely,

Shan Zha

Guest Editor

PLOS Genetics

Peter McKinnon

Section Editor

PLOS Genetics

Comments from the reviewers (if applicable):

Congratulations! We are please to accept the manuscript for publication. Please address the minor changes on UVB source and technical deatils brought by the reviewers.

Reviewer's Responses to Questions

**Comments to the Authors:**

Reviewer #1: Revised manuscript: PGENETICS-D-22-00803R1

Reviewer 1.

I have carefully looked at the revised manuscript particularly the changes and additions made by the authors in response to the comments by both referees.

Comment 1. The point that PARP1-DNA-bound complexes, due to PARP inhibitor Olaparib might interfere with replication complicating the interpretation is addressed by the authors in their response. Although Olaparib-trapped PARP1 DNA complexes are to some extent dynamic, this does not completely rule out that they still interfere with replication. However, I am satisfied with this response.

Comment 2. The mapping of the interaction domains between SHPRH are PARP1, which was only partially carried out, has been completed in revision and has revealed additional portions by which these proteins appear to interact as demonstrated in the new data in Fig 3b and supplementary Fig S9 and S10a-c and in the text at several places. This implies that the proteins have multiple interaction domains which may be relevant as the authors speculate that the proteins constantly interact with each other (albeit very likely in a dynamic manner) in a DNA-damage-independent fashion. I am happy with the new data.

Comment 3. In response the authors have added a number of data after UV exposure which permit a more general statement suggesting that replication slowing and the formation of reversed forks are global responses to genotoxic treatments and this process requires PARP1/DNA translocases recruitment to damaged forks. These data are included in supplementary Fig S5 and Fig S12 and extensively discussed. A minor comment: the experimental conditions used for the added UV exposure experiments have not been described in the Materials and Methods section. Was a UV-B source used for the UV exposure studies? When UV-C would have been used the dose would be much too high. This should be specified in the text as well. This referee is satisfied by the addition of these data.

Comment 4. This comment concerned the proper controls for some of the immunoblots. These blots have been replaced in Fig 6a and Fig 8b-e, which takes away the concern.

Comment 5. The overstatement that their findings “… explain the mechanism of PARP1 in fork progression restraint upon replication stress.” has been removed, which properly addresses my comment.

Overall, with the minor point on the UV source and experimental details I can recommend acceptation.

Reviewer #2: The authors have addressed all my concerns.

**Have all data underlying the figures and results presented in the manuscript been provided?**

Reviewer #1: Yes

Reviewer #2: Yes

PLOS authors have the option to publish the peer review history of their article (what does this mean?). If published, this will include your full peer review and any attached files.

Reviewer #1: No

Reviewer #2: No

**Data Deposition**

http://datadryad.org/submit?journalID=pgenetics&manu=PGENETICS-D-22-00803R1

**Press Queries**

---

## [Editor Report · Acceptance letter]

6 Dec 2022

PGENETICS-D-22-00803R1 

PARP1 recruits DNA translocases to restrain DNA replication and facilitate DNA repair 

Dear Dr Liaw, 

We are pleased to inform you that your manuscript entitled "PARP1 recruits DNA translocases to restrain DNA replication and facilitate DNA repair" has been formally accepted for publication in PLOS Genetics! Your manuscript is now with our production department and you will be notified of the publication date in due course.

With kind regards,

Olena Szabo

PLOS Genetics

On behalf of:
